# Investigating the Effects of Diet-Induced Prediabetes on Skeletal Muscle Strength in Male Sprague Dawley Rats

**DOI:** 10.3390/ijms25074076

**Published:** 2024-04-06

**Authors:** Mandlakazi Dlamini, Andile Khathi

**Affiliations:** Department of Human Physiology, School of Laboratory Medicine, Medical Sciences, College of Health Sciences, University of KwaZulu-Natal, Durban X54001, South Africa; 218004674@stu.ukzn.ac.za

**Keywords:** four-limb grip strength test, muscle strength, prediabetes, high-fat high carbohydrate diet, insulin resistance, hyperglycemia, glycated hemoglobin, lipid peroxidation

## Abstract

Type 2 diabetes mellitus, a condition preceded by prediabetes, is documented to compromise skeletal muscle health, consequently affecting skeletal muscle structure, strength, and glucose homeostasis. A disturbance in skeletal muscle functional capacity has been demonstrated to induce insulin resistance and hyperglycemia. However, the modifications in skeletal muscle function in the prediabetic state are not well elucidated. Hence, this study investigated the effects of diet-induced prediabetes on skeletal muscle strength in a prediabetic model. Male Sprague Dawley rats were randomly assigned to one of the two groups (*n* = 6 per group; six prediabetic (PD) and six non-pre-diabetic (NPD)). The PD group (*n* = 6) was induced with prediabetes for 20 weeks. The diet that was used to induce prediabetes consisted of fats (30% Kcal/g), proteins (15% Kcal/g), and carbohydrates (55% Kcal/g). In addition to the diet, the experimental animals (*n* = 6) were supplied with drinking water that was supplemented with 15% fructose. The control group (*n* = 6) was allowed access to normal rat chow, consisting of 35% carbohydrates, 30% protein, 15% fats, and 20% other components, as well as ordinary tap water. At the end of week 20, the experimental animals were diagnosed with prediabetes using the American Diabetes Association (ADA) prediabetes impaired fasting blood glucose criteria (5.6–6.9 mmol/L). Upon prediabetes diagnosis, the animals were subjected to a four-limb grip strength test to assess skeletal muscle strength at week 20. After the grip strength test was conducted, the animals were euthanized for blood and tissue collection to analyze glycated hemoglobin (HbA1c), plasma insulin, and insulin resistance using the homeostatic model of insulin resistance (HOMA-IR) index and malondialdehyde (MDA) concentration. Correlation analysis was performed to examine the associations of skeletal muscle strength with HOMA-IR, plasma glucose, HbA1c, and MDA concentration. The results demonstrated increased HbA1c, FBG, insulin, HOMA-IR, and MDA concentrations in the PD group compared to the NPD group. Grip strength was reduced in the PD group compared to the NPD group. Grip strength was negatively correlated with HbA1c, plasma glucose, HOMA-IR, and MDA concentration in the PD group. These observations suggest that diet-induced prediabetes compromises muscle function, which may contribute to increased levels of sedentary behavior during prediabetes progression, and this may contribute to the development of hyperglycemia in T2DM.

## 1. Introduction

Skeletal muscle is the largest organ, making up approximately 40% of human body weight [1]. It plays a prominent role in ambulation, posture, glucose homeostasis, and metabolism [2]. Satellite cells are an integral part of skeletal muscle structure, located between the skeletal myofiber’s basal lamina and plasma membrane, and are responsible for maintaining skeletal muscle health [3]. Satellite cells are usually mitotically latent in adult skeletal muscle and only become functional upon physiological stimulation [2]. Stimulation of the satellite cells results in satellite cell activation, proliferation, and terminal differentiation [4]. This, in turn, promotes the formation of new myofibers or the addition of satellite cell nuclei to existing myofibers, thereby promoting muscle growth and regeneration [4]. Myofiber type, size, and quantity are proposed to play a distinct role in skeletal muscle strength [5]. Skeletal muscle fibers are categorized into two fiber types, namely “slow-twitch” and “fast-twitch” (type II) muscle fibers [6]. Type I is the preferable muscle fiber type involved in muscle strength since it is associated with sustainable energy production [6]. Defects in the satellite cell population and myofibers can contribute to the pathogenesis of muscle disease [7].

Type 2 diabetes mellitus (T2DM) is observed to compromise skeletal muscle structure and muscle strength by altering the quantity and function of satellite cells and myofiber type proportion, respectively [8]. Previous studies have documented compromised skeletal muscle regeneration capacity in T2D rats resulting from reduced satellite cell activation and proliferative capacity observed in the T2D state [9]. The hyperglycemia and lipotoxicity associated with T2DM are proposed to induce delayed myofiber maturation, thereby affecting muscle regeneration capacity [10]. T2DM is also associated with reduced muscle contractile activity due to the reduced proportion of type II muscle fibers observed in T2D rats [11]. The reduction in skeletal muscle fiber observed in T2D rats reduces muscle fiber number and size, affecting muscle mass and strength [11]. Other studies have documented diminished forelimb grip strength in T2D rats [11] resulting from the reduced contractile force observed in T2D conditions [12]. The onset of T2DM, however, is often preceded by prediabetes [13].

Prediabetes is an intermediate state of hyperglycemia, with blood glucose parameters above the normal range but below the diabetes threshold [14]. It is a condition characterized by impaired fasting glucose (IFG), impaired glucose tolerance, and elevated glycated hemoglobin (HbA1c) concentrations [15]. The global prevalence rate of prediabetes in 2017 was estimated to be 352 million (7.3%) in the adult population, and this number is projected to increase to 587 million (8.3%) individuals by 2045 [16]. Previous studies have documented that the insulin resistance and β-cell dysfunction associated with prediabetes play a role in the onset of micro- and macrovascular diabetic complications [17,18].

Diabetic myopathy is one of the T2DM-associated complications resulting from long-standing, poorly controlled T2DM, consequently inducing skeletal muscle inflammation, ischemia, and compromised muscle strength [12,19,20,21,22]. The literature predominantly documents T2DM-associated changes in skeletal muscle strength, while studies documenting prediabetes-associated changes in skeletal muscle strength are very few. Hence, this study aims to investigate changes in skeletal muscle strength using a high-fat, high-carbohydrate (HFHC) diet-induced prediabetic model, mainly focusing on the association between skeletal muscle grip strength and prediabetes-associated metabolic derangements such as hyperglycemia, elevated HbA1c concentrations, skeletal muscle fat infiltration, and insulin resistance. The rationale behind using the HFHC diet is the similar metabolic manifestations of the human prediabetic condition [23]. Research on prediabetes is essential because establishing diabetic biomarkers during the prediabetic state can potentially prevent the development of T2DM, as prediabetes is a reversible condition if detected in the early stages [24].

## 2. Results

### 2.1. Glycated Hemoglobin

Glycated hemoglobin concentrations were measured in the non-prediabetic (NPD) group (*n* = 6) and prediabetic (PD) group (*n* = 6) at the end of the experimental period. It was observed (Figure 1) that the concentration of glycated hemoglobin was significantly (*p* = 0.0097) higher in the PD group than in the NPD group (Figure 1).

### 2.2. Fasting Plasma Glucose, Insulin, and HOMA-IR

Fasting plasma glucose concentrations were measured in the non-prediabetic (NPD) group (*n* = 6) and prediabetic (PD) group at the end of the experimental period. The concentrations of fasting glucose (*p* = 0.0053) and insulin (*p* < 0.0001) were observed to be significantly higher in the PD group than in the NDP group. Furthermore, the PD group had a significantly (*p* < 0.0001) higher HOMA-IR value compared to the NPD group, which was within the range of significant insulin resistance (>2.9), whereas the NPD group’s HOMA-IR value was within the insulin-sensitive range (1.0) (Table 1).

### 2.3. Oral Glucose Tolerance Test (OGTT)

The OGTT and area under curve (AUC) were measured in the non-prediabetic (NPD) group (*n* = 6) and prediabetic (PD) group (*n* = 6) after the experimental period. It was evident (Figure 2) that the concentration of FBG in the PD group was significantly (*p* = 0.0016) higher at time 0 than in the NPD group. FBG concentrations in the PD group were significantly higher at 30 min (*p* = 0.0014) and 120 min (*p* < 0.0001) after glucose loading compared to the NPD group. Furthermore, the AUC for the PD group was significantly (*p* = 0.0029) higher than that of the NPD group.

### 2.4. Four-Limb Grip Strength Test

The non-prediabetic (NPD) and prediabetic (PD) groups (*n* = 6) were subjected to a reverse hanging test (Figure 3a) at the end of week 20. The minimum holding impulse (body mass (g) × time (s)) was calculated for the NPD and PD groups to correct the negative effects of body mass on hang time (Figure 3b). It was evident that grip strength was significantly lower in the reverse hang test (*p* = 0.005) (Figure 3a). The reverse hanging test’s minimum holding impulse (Figure 3b) was also significantly lower in the PD groups (*p* = 0.0088).

### 2.5. Malondialdehyde (MDA) Measurement

The concentration of MDA was measured in the non-prediabetic (NPD) group (*n* = 6) and prediabetic (PD) group (*n* = 6) after the experimental period. It was evident (Figure 4) that the concentration of MDA in the PD group was significantly (*p* = < 0.0001) higher than in the NPD group.

### 2.6. Correlation between HOMA-IR, Hb1Ac, Plasma Glucose, MDA Concentration, and Grip Strength

Pearson’s correlation analyses were performed between values for HOMA-IR, glycated hemoglobin, plasma glucose, MDA concentration, and grip strength in the non-prediabetic (NPD) group (*n* = 6) and prediabetic (PD) group (*n* = 6) after the experimental period (Table 2). It was evident (Table 2) that there was a negative correlation between grip strength (r = −0.94; *p* = 0.01) and HbA1c in the PD group. Grip strength (r = −0.98; *p* = 0.02) was negatively correlated with insulin resistance in the PD group. Furthermore, plasma glucose was negatively correlated with grip strength (r = −0.86; *p* = 0.03) in the PD group. MDA concentrations were negatively correlated with reverse grip strength (r = −0.86; *p* = 0.03) in the PD group

## 3. Discussion

Skeletal muscle comprises cells or fibers that produce body force and movement and contribute to maintaining whole-body homeostasis [25,26]. The skeletal muscle satellite cells are suggested to play a prominent role in muscle fiber maintenance, repair, and remodeling, ultimately maintaining skeletal muscle plasticity and muscle functional capacity [4]. Previous studies have documented that T2DM, a condition often preceded by prediabetes, considerably affects muscle health, consequently affecting skeletal muscle structure, strength, and glucose homeostasis [8,27]. The animals in this study were induced with prediabetes in our laboratory using a well-established protocol comprising chronic consumption of a high-fat high-carbohydrate (HFHC) diet [23,28,29]. The studies that used this protocol document that there is an onset of T2DM complications, such as cardiovascular disorders, immune dysregulation, and renal failure, in the prediabetic state [23,28,29]. Previous studies have predominantly reported on T2DM-associated changes in the skeletal muscle, but prediabetes-associated skeletal muscle changes are not well elucidated [12,30]. Hence, this study investigated the effects of diet-induced prediabetes on skeletal muscle strength and structure. Moreover, the study investigated the association of grip strength with glycated hemoglobin, hyperglycemia, skeletal muscle fat infiltration, and insulin resistance in the prediabetic state.

Prediabetes is suggested to be an asymptomatic metabolic disorder that is associated with early insulin resistance and blood glucose levels that are not within the homeostatic range [24]. Elevated glucose concentrations stimulate insulin release and circulation, promoting peripheral tissue glucose absorption [14]. This accounts for the plasma glucose concentrations that are within the physiological homeostatic range in glucose-tolerant individuals [31]. Furthermore, glucose levels are not continuously high enough in glucose-tolerant individuals to induce increased glycation of hemoglobin and insulin resistance [32]. However, in the prediabetic state, there is elevated glucose concentration, insulin secretion, HbA1c, and preliminary insulin resistance compared to glucose-tolerant individuals [14]. In prediabetic conditions, plasma insulin cannot trigger a response in insulin-dependent peripheral tissues such as the skeletal muscle [17]. Hence, the increase in glucose concentrations in the blood plasma results in an elevated glycation rate of hemoglobin [33]. Consequently, pancreatic β-cells compensate for the increased blood glucose concentrations by releasing a substantial amount of insulin to decrease the heightened glucose concentrations, leading to compensatory hyperinsulinemia [33].

Impaired glucose tolerance (IGT), along with impaired fasting glucose (IFG), are documented to contribute to moderate insulin resistance in insulin-dependent tissues [17]. Increased blood glucose concentration is observed during the postprandial state of glucose-tolerant individuals, stimulating insulin release to promote glycogenesis and inhibit glycogenolysis [34]. Hence, plasma glucose levels are maintained within the normal physiological range, followed by plasma insulin levels returning to the homeostatic range [35]. However, there is increased endogenous glucose production in the PD state before food consumption and in the postprandial state [36] because of the impaired insulin-induced peripheral glucose uptake in insulin-dependent tissue of prediabetic individuals [23]. This accounts for the high insulin, postprandial glucose levels, HOMA-IR, and fasting plasma glucose in prediabetic patients compared to glucose-tolerant patients [17,37]. In this study, the postprandial glucose concentration at 120 min, AUC, HbA1c, and HOMAR-IR values were significantly higher in the PD group compared to the NDP group. These findings correspond with previous findings showing significantly higher plasma glucose, insulin, HbA1c, 2 h postprandial glucose levels, and HOMA-IR values in prediabetic patients compared to NPD patients [33,35,38,39]. In the PD group, the increased plasma insulin, impaired fasting glucose, impaired glucose tolerance, HbA1c, and HOMA-IR values in the insulin resistance range may suggest some insulin resistance from peripheral tissue against glucose uptake.

This study further validates that chronic consumption of the HFHC diet results in the development of prediabetes, as seen by impaired fasting glucose, impaired glucose tolerance, elevated plasma insulin, HbA1c, and HOMA-IR values within the range of insulin resistance, which suggests that the body’s ability to use glucose in insulin-dependent tissues has been affected [40]. High intake of dietary fats, associated with elevated triacylglycerides, may have also contributed to the development of insulin resistance as it has been observed that increased exposure of triacylglycerides to insulin-dependent peripheral tissues induces insulin resistance [17]. Hence, this potentially resulted in the pancreatic beta (β) cells over-secreting insulin to compensate for the increased plasma glucose concentration in the insulin-resistant state [39]. Skeletal muscle structure and strength are documented to be implicated in T2DM-associated metabolic derangements such as hyperglycemia, insulin resistance, oxidative stress, and intramuscular fat infiltration [30,41,42].

This study showed a significant decrease in grip strength in the PD groups compared to the NPD groups. These results correspond with previous findings demonstrating reduced grip strength in T2DM rats, T2DM individuals, and prediabetic individuals [11,43,44,45]. Reduced grip strength is documented to be associated with muscle fiber type composition [11]. Grip strength is suggested to be chiefly dependent on type II fiber cross-sectional area (CSA) [11]. Similarly, a positive correlation was found between type II fiber CSA in the vastus lateralis (VL) muscle and the maximum extension strength of leg muscles in T2DM patients [46]. In another study, forelimb grip strength and the fiber CSA of fast glycolytic TA muscles were reduced in type 2 diabetic mice, with fast-type muscle mass also decreasing [47]. Based on the above observations, the reduced grip strength in this study’s PD group could be attributed to fast-type muscle atrophy accompanied by type IIb fiber atrophy [48]. Inflammation associated with T2DM and prediabetes is also proposed to affect grip strength by impacting muscle fibers [11,12,49]. High levels of TNF-α and interleukin (IL)-6 were documented in T2DM individuals with excessive loss of skeletal muscle mass, and low grip strength was associated with the high levels of TNF-α/IL-6 [50,51].

Takada et al. [11] propose that grip-associated proteins’ non-enzymatic glycosylation (glycation) may attenuate grip strength because the elevated glycosylated hemoglobin in T2D rats correlates negatively with grip strength. A significant negative correlation was also found between HbA1c and grip strength in patients with T2DM [46]. This study documented a significant negative correlation between HbA1c and the grip strength of PD groups, which is in agreement with previous studies [11,46]. Skeletal muscle myosin from T2DM patients was more glycosylated and had lower calcium ATPase activity than the control myosin of healthy subjects [52]. Similarly, rat single muscle fibers exposed to glucose-6-phosphate presented with significantly reduced maximum calcium-activated force per CSA and magnesium-activated ATPase activity [53]. Moreover, rat skeletal muscle myosin was incubated with glucose and analyzed for structural and functional changes [54]. Upon analysis, glycation-related structural alterations were associated with a significant reduction in the in vitro mortality speed, suggesting a structure-related decline in myosin mechanics in response to glucose exposure [54]. These findings potentially account for the PD group’s negative correlation between HbA1c and grip strength.

Low muscle strength is reported as a potential marker for impaired skeletal muscle glucose disposal, which is closely associated with whole-body insulin resistance [55]. Insulin resistance is proposed to contribute to the impairment of muscle strength and performance as it plays a substantial role in regulating muscle protein metabolism. Furthermore, insulin also promotes mitochondrial protein synthesis, a crucial process required for maintaining mitochondria proteins and their functional activity [56]. In T2DM patients, skeletal muscle mitochondrial protein synthesis and cytochrome C oxidase, a prominent enzyme for ATP production, have been reported to be unresponsive to insulin treatment [57]. These structural and functional changes in the skeletal muscle induced by defective insulin action are suggested to be associated with muscle weakness and reduced endurance capacity, which in turn may exacerbate insulin resistance [55]. As a result, this study documents a significant negative correlation between insulin resistance and muscle strength, consistent with previous findings that observed a significant negative correlation between insulin resistance and muscle strength [55].

The hyperglycemia observed in T2DM is suggested to play a role in reduced muscle strength associated with T2DM [58]. Diet-induced T2DM and prediabetes-associated hyperglycemia are proposed to contribute to generating reactive oxygen species (ROS), resulting in mitochondrial dysfunction and compromised muscle function [59,60,61,62]. Mitochondria serve as a vital energy source in the preservation of normal skeletal muscle function [59]. Chronically elevated plasma glucose concentrations result in an accumulation of free fatty acids (FFA), leading to a high saturation of energy substrates in the muscle and the development of intramyocellular lipid accumulation [63]. This results in ROS production through increased mitochondrial uncoupling [64] and increased NAD(P)H oxidase enzyme activity [65]. An increase in ROS production can result in disrupted mitochondrial biogenesis and mitochondrial DNA replication, which promotes mitochondrial dysfunction.

Consequently, fatty acid oxidation is downregulated, heightening skeletal muscle lipid deposition [66]. This creates a vicious cycle in which elevated intramuscular lipids stimulate the formation of lipid peroxides known to damage the mitochondria [66]. This study showed a significant negative correlation between plasma glucose concentrations and muscle grip strength, corroborating previous findings that documented a significant negative correlation between plasma glucose and grip strength [11]. The negative correlation between muscle strength and plasma glucose could be attributed to the mitochondrial dysfunction associated with hyperglycemia in the T2DM and prediabetic state [11,55,56,57,58].

Inter- and intramuscular adipose tissue (IMAT) is an ectopic fat depot associated with poor metabolic [67] and muscle health outcomes [68,69]. T2DM individuals are reported to present with high amounts of IMAT compared to lean controls, and IMAT is proposed to be associated with insulin sensitivity [70]. In addition to influencing insulin sensitivity, several studies have documented that IMAT is an independent predictor of physical function [68]. Considering that IMAT consists of non-contractile tissue, skeletal muscle fat infiltration negatively affects the properties of skeletal muscle by disrupting the muscle’s capacity to generate force, displacing contractile materials, changing skeletal muscle elastic properties, and consequentially affecting skeletal muscle contraction dynamics [71], which compromises the efficient transmission of contractile forces [72]. Studies document that prediabetic men have lower insulin sensitivity compared to women, which possibly results from the lower intramuscular triglyceride oxidation and turnover rate with subsequent accumulation of intramuscular triglycerides and physical dysfunction [73]. 

The results of this study illustrate significantly elevated MDA (marker of lipid peroxidation) concentrations in the PD group compared to the NDP group. A negative correlation was also observed between MDA concentration and grip strength in the PD group. These results correspond with previous findings that demonstrate increased skeletal muscle lipid peroxidation due to hyperglycemia and a negative association between lipid accumulation and muscle strength [70,73]. Hence, the reduced muscle strength observed in this study’s PD group could be attributed to the skeletal muscle fat infiltration.

It was evident that reduced grip strength is present in the PD state, indicating compromised skeletal muscle function. In addition, skeletal muscle glucose homeostasis may have been compromised due to high plasma glucose, insulin, lipid peroxidation, HbA1c, and HOMA-IR index values within the insulin-resistant range in the PD state. Furthermore, HbA1c, plasma glucose, MDA concentration, and insulin resistance were negatively correlated with muscle grip strength in the PD state. Compromised muscle function may contribute to increased levels of sedentary behaviors during prediabetes progression, and this may contribute to the development of hyperglycemia and insulin resistance in T2DM [74].

## 4. Materials and Methods

### 4.1. Animals and Housing

This study employed male Sprague Dawley rats (150–180 g) bred and housed at the University of KwaZulu-Natal Biomedical Research Unit (BRU). The animals were kept under standard laboratory conditions, which included a constant temperature of 22 ± 2 °C, carbon dioxide (CO_2_) content of <5000 p.m., a relative humidity of 55 ± 5%, and illumination (12 h light/dark cycle, lights on at 07h00). The noise level was maintained at less than 65 decibels. The animals were allowed access to food and fluids ad libitum. The Animal Research Ethics Committee of the University of KwaZulu-Natal (ETHICS#: AREC/00004187/2022) approved all animal experimentation. The animals were allowed to acclimatize to their new environment for one week while consuming standard rat chow and tap water before exposure to the experimental diets [23]. The animal care procedures followed the University of KwaZulu-Natal’s institutional guidelines for animal care.

### 4.2. Induction of Prediabetes

For an experimental period of 20 weeks, rats were randomly divided into two groups (*n* = 6 per group) and fed their respective diets. Experimental prediabetes was induced in the animals using a protocol previously described by Luvuno et al. [23]. To induce prediabetes, one group was fed a high-fat high-carbohydrate (HFHC) diet supplemented with 15% fructose-enriched water (AVI Products (Pty) Ltd., Waterfall, South Africa), whereas the other group was fed standard rat chow and supplied with tap water. The diet that was used to induce prediabetes consisted of fats (30% Kcal/g), proteins (15% Kcal/g), and carbohydrates (55% Kcal/g). In contrast, the other group was fed standard rat chow and supplied with tap water. The normal rat chow comprised carbohydrates (65% Kcal/g), protein (25% Kcal/g), and fats (15% Kcal/g). The animals were evaluated for prediabetes after 20 weeks using the American Diabetes Association (ADA) criteria [75]. Animals with fasting blood glucose (FBG) concentrations within the range of 5.6 to 6.9 mmol/L, impaired glucose tolerance (IGT) with plasma glucose levels of 7.8 to 11.0 mmol/L 2 h postprandial, and HbAc1 concentrations of 5.7 to 6.4% were regarded as prediabetic. The animals that were fed the standard diet were also tested at week 20 to confirm normoglycemia.

### 4.3. Experimental Design

This study comprised two groups, namely a non-prediabetic (NPD) group and a prediabetic (PD) group (*n* = 6 in each group). The NPD group consisted of animals that consumed the standard rat chow and water for 20 weeks and did not have prediabetes, while the PD group consisted of animals that consumed the HFHC diet supplemented with 15% fructose for the same number of weeks, and which were diagnosed with prediabetes. At the end of week 20, the PD and NPD groups were subjected to a four-limb grip strength test to assess the effect of prediabetes on skeletal muscle strength. At the end of the experimental duration, blood and skeletal muscle tissue collections were performed.

### 4.4. Oral Glucose Tolerance (OGT) Response

At week 20, an oral glucose tolerance test (OGTT) was conducted following glucose loading to determine the glucose tolerance response of animals subjected to the chronic ingestion of the HFHC diet. The OGT responses were monitored in the animals according to a well-established protocol [76]. Briefly, after a 12 h fast, glucose levels were measured (time: 0 min) in all animals. The animals were loaded with glucose (glucose: 0.86 g/kg) through an oral gavage (18-gauge curved 38 mm long gavage needle with a 21/4 mm ball end). Blood was collected using the tail prick method to measure glucose concentration [77]. Glucose concentrations were measured using a Viva check glucometer (Hangzhou, China). The glucose concentrations were measured at 15, 30, 60, and 120 min following glucose loading.

### 4.5. Four-Limb Grip Strength Test

A reverse hanging test was performed to evaluate grip strength in the NDP and PD groups (Figure 5) [11]. A wire mesh grid (dimensions: 80 × 50 cm; wire mesh spacing: 12 mm; wire thickness: 1.5 mm) was used. The grid was set at a height of about 60 cm, and soft bedding was placed underneath to protect the rats should they fall off the grid [78]. In the reverse hanging test, each test animal was allowed to grasp the mesh grid with all paws in a face-up position. The grip-holding time was measured until the grip was released. The procedure was repeated twice by the same person, and the longest hanging time was recorded [11]. Animals that immediately jumped off from the grid were retested. The minimum holding impulse (body mass(g) × time(s)) was calculated for the NPD and PD groups to correct the negative effects of body mass on hang time [79].

### 4.6. Tissue and Blood Collection

At the end of the experimental period, all animals were euthanized using a guillotine. Trunk blood was collected into heparinized containers. The blood was centrifuged (Eppendorf centrifuge 5403, LGBW, Germany) for 15 min at 4 °C (503 g). Plasma was isolated from blood and stored in a Bio Ultra Freezer (Labotec, Umhlanga, South Africa) at −80 °C until biochemical analysis as previously described by Luvuno et al. [23]. The isolated plasma was used to assess HbA1c concentrations and plasma insulin concentrations in the NPD and PD groups. The gastrocnemius skeletal muscle tissues were harvested from the NDP and PD groups and cut in half, rinsed with cold standard saline solution, and snap-frozen in liquid nitrogen before storage either in a Bio Ultra Freezer at −80 °C for biochemical analysis or in formalin for histological analysis [23].

### 4.7. HOMA-IR Index

Insulin resistance was calculated from fasting blood glucose and insulin levels using the homeostatic model assessment (HOMA) [17]. The HOMA-IR index was calculated using the HOMA2 Calculator v2.2.3 program [80]. Insulin sensitivity was indicated by values <1.0, early insulin resistance by values >1.9, and significant insulin resistance by values >2.9.

### 4.8. Biochemical Analysis

According to the manufacturer’s instructions, the glycated hemoglobin (HbA1c) and plasma insulin were measured using an HbA1c ELISA kit (Elabscience and Biotechnology, Wuhan, China). Micro-ELISA plates were coated with antibodies as part of the standard experimental protocol in the ELISA kits. The plasma samples were pipetted into the appropriate wells, followed by immediate addition of the appropriate biotinylated detection antibody (50 μL). The samples were then incubated for 45 min at 37 °C, after which the unbound components were washed away with the supplied wash buffer. After washing, the wells were filled with 100 μL of avidin–horseradish peroxidase (HRP), which was incubated at 37 °C for 30 min. After removing the unattached components with a second wash, the substrate reagent (90 μL) was applied to the wells. This was followed by a 15 min incubation period at 37 °C. Finally, a stop solution (50 μL) was applied to the micro-wells to stop the reaction and allow for appropriate measurements. Optical density at 450 nm was determined using a nano-spectrophotometer (BMG Labtech, Ortenburg, Germany). The samples’ glycated hemoglobin and plasma insulin concentrations were extrapolated from their respective standard curves. Lipid peroxidation was estimated by determining the concentration of MDA in the skeletal muscle tissue homogenate according to a previously described protocol [81].

### 4.9. Statistical Analysis

The mean and the standard error of the mean (SEM) were used to represent the data. Statistical comparisons were performed with Graph Pad InStat software (version 8.00, Graph Pad Software, Inc., San Diego, CA, USA). The Shapiro–Wilk normality test was used to assess data distribution and the *p*-value was greater than 0.05; hence, a normal data distribution was assumed. The Student *t*-test was used to determine statistical differences between two independent groups. Pearson’s correlation test was used to assess the association of muscle strength with HOMA-IR, glycated HbA1, plasma glucose, and MDA concentration in the NDP and PD groups. A value of *p* ˂ 0.05 was considered statistically significant. A coefficient value between ±0.70 and ±1.0 was considered strong.

## 5. Limitations and Future Recommendations

The primary limitation of this study is the lack of extensive mechanistic research to support the findings of the skeletal muscle strength behavioral study due to logistical problems and a shortage of lab supplies. Therefore, future studies must investigate the impact of diet-induced prediabetes on skeletal muscle structure, particularly the skeletal muscle morphological changes in skeletal muscle satellite cells, myofiber type, and composition, as these biomarkers are collectively suggested to be associated with skeletal muscle functional capacity. Molecular biomarkers, such as protein expression of myogenic regulatory factors of skeletal muscle satellite cells and gene expression of myofiber type, must also be assessed because they are suggested to influence skeletal muscle structure and function. Inflammatory markers associated with diet-induced prediabetes also need to be evaluated, as T2DM-induced inflammation is documented to contribute to compromised skeletal muscle function.

## 6. Conclusions

This study showed evidence of prediabetes-related metabolic derangements such as insulin resistance and elevated plasma glucose and HbA1c and MDA concentrations. The effects on skeletal muscle functional capacity associated with diet-induced prediabetes, including reduced muscle grip strength, are documented to contribute to the development of reduced muscle grip strength during the prediabetic state. Hence, these findings suggest that there is an onset of decreased muscle strength in the prediabetic state, which can potentially contribute to the development of diabetic myopathy in progressive prediabetes and T2DM. The functional changes in the skeletal muscle induced by defective insulin action, observed in the prediabetic state, are suggested to be associated with impaired muscle functional capacity, which in turn may exacerbate insulin resistance and promote the development of T2DM.

## Figures and Tables

**Figure 1 ijms-25-04076-f001:**
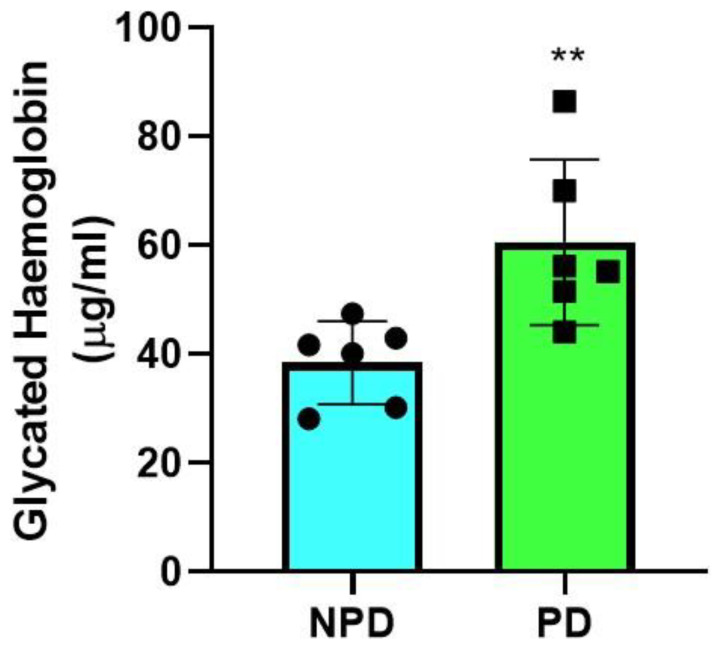
Glycated hemoglobin concentrations in the non-prediabetic (NPD) and prediabetic (PD) groups (*n* = 6 per group). The values are depicted as mean ± SEM. ** = *p* ˂ 0.05 when compared to the NPD group. Blue bar represents NPD group and green bar represents PD group.

**Figure 2 ijms-25-04076-f002:**
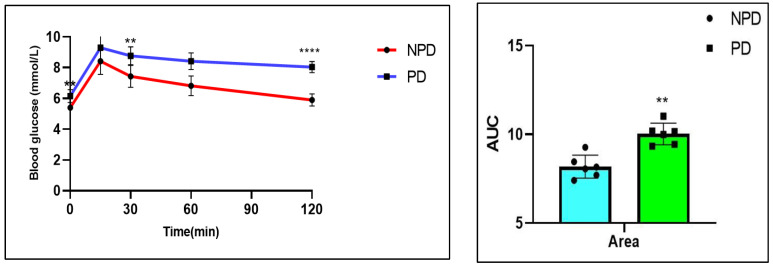
The OGTT response and AUC values in the non-prediabetic (NPD) group and prediabetic group (PD) (*n* = 6 per group). Values are depicted as mean ± SEM. ** = *p* < 0.01, **** = *p* < 0.0001 when compared to the NPD group. For area under the curve bar graph, the blue bar represents NPD group and green bar represents PD group.

**Figure 3 ijms-25-04076-f003:**
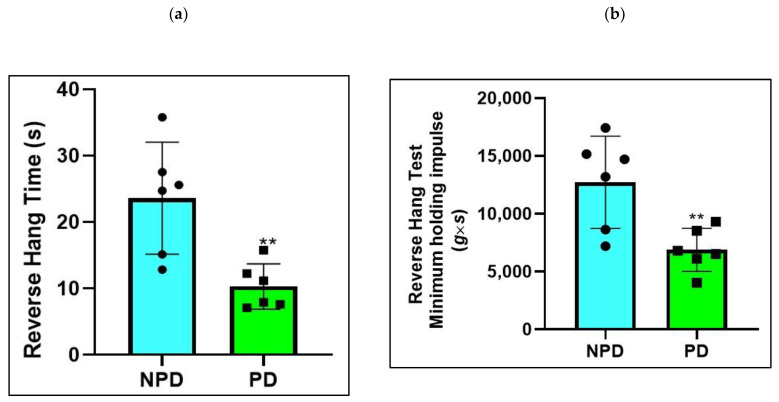
Hanging time in the reverse hanging test of non-prediabetic groups (NPD) and prediabetic groups (PD) (*n* = 6) (**a**). Minimum holding impulse for the reverse hang test of non-prediabetic (NPD) and prediabetic (PD) groups (*n* = 6) (**b**). The values are depicted as mean ± SEM. ** = *p* ˂ 0.05. Blue bar represents NPD groups and green bar represents PD groups.

**Figure 4 ijms-25-04076-f004:**
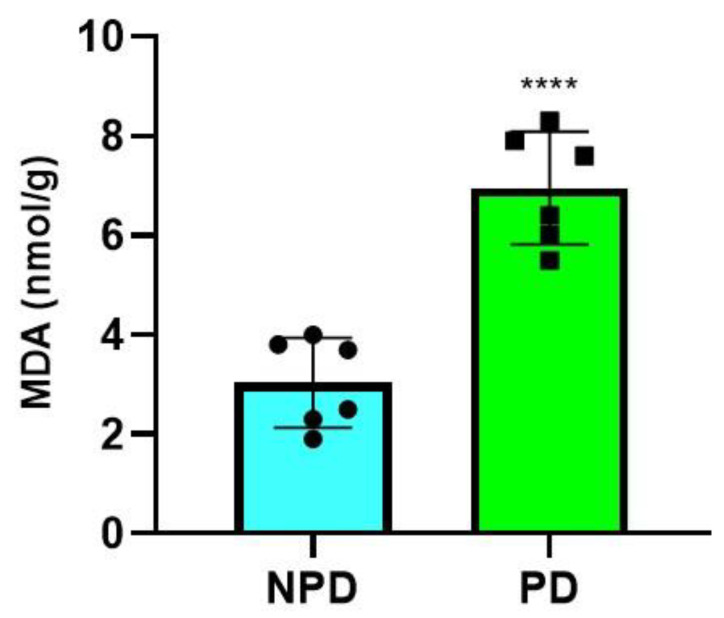
MDA concentrations in the non-prediabetic (NPD) and prediabetic (PD) groups (*n* = 6 per group). The values are depicted as mean ± SEM. **** = *p* ˂ 0.0001. Blue bar represents NPD group and green bar represents PD group.

**Figure 5 ijms-25-04076-f005:**
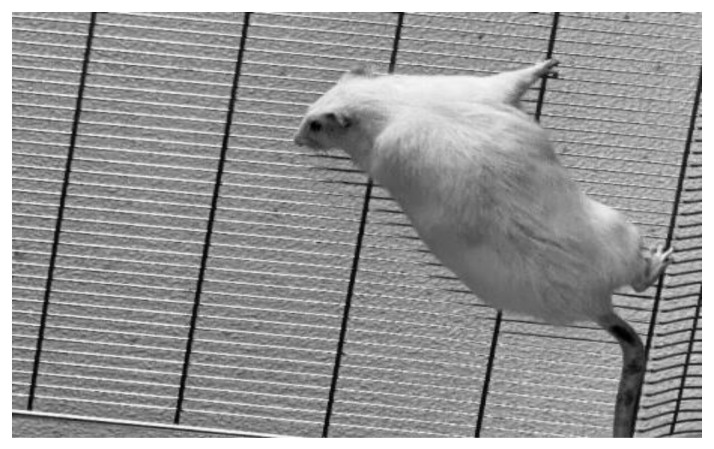
Photograph of an animal in the reverse hanging test.

**Table 1 ijms-25-04076-t001:** Concentrations of plasma glucose, insulin, and HOMA-IR indices in the non-prediabetic (NPD) and prediabetic group (PD) (*n* = 6 per group). The values are depicted as mean ± SEM. ** = *p* < 0.05; **** = *p* < 0.0001 when compared to the NPD group.

Groups (*n* = 6)	Plasma Glucose (mmol/L)	Plasma Insulin (pmol/L)	HOMA-IR Values
NPD	5.4 ± 0.08	30.42 ± 3.26	0.59 ± 0.06
PD	6.15 ± 0.17 **	322.80 ± 17.64 ****	6.16 ± 0.40 ****

**Table 2 ijms-25-04076-t002:** Pearson’s rank correlation between values for HOMA-IR, HbA1c, plasma glucose, and grip strength in the non-prediabetic (NPD) and prediabetic (PD) group (*n* = 6 per group). * = *p* < 0.05; ** = *p* < 0.01.

Metabolic Parameters		Reverse Hang Test Minimum Holding Impulse
		(g × s)
HbA1c	NPD	r = 0.33
		*p* = 0.67
	PD	r = −0.94
		*p* = 0.005 **
HOMA-IR	NPD	r = 0.10
		*p* = 0.85
	PD	r = −0.98
		*p* = 0.02 *
Plasma glucose	NPD	r = 0.04
		*p* = 0.46
	PD	r = −0.86
		*p* = 0.03 *
MDA	NPD	r = 0.24
		*p* = 0.65
	PD	r = −0.86
		*p* = 0.03 *

## Data Availability

The data presented in this study are available on request from the corresponding author (Appendix A).

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
