# Peer review of "Investigating the Effects of Diet-Induced Prediabetes on Skeletal Muscle Strength in Male Sprague Dawley Rats"

_ijms, 2024, doi:10.3390/ijms25074076_

Round 1

Reviewer 1 Report

Comments and Suggestions for Authors

Comments:

Abstract:

1. In the abstract, authors are suggested to include the specific diet composition used in the induction of pre-diabetes. Authors should add brief details on the high-fat high-carbohydrate diet such as the percentage of macronutrients and sources of fat and carbohydrates.

Introduction:

2. The introduction provides a comprehensive overview of satellite cells and their role in skeletal muscle health. However, it lacks a clear statement of the research gaps and the specific aim of the study. Authors are suggested to include the gap in knowledge and aims to address the effects of diet-induced pre-diabetes on skeletal muscle structure and strength.

Materials and Methods:

3. The methods section lacks information on the duration of the pre-diabetic state induction. Authors should specify the time for the rats to develop pre-diabetes under the experimental conditions.

4. The criteria used for diagnosing pre-diabetes in the rats should be mentioned.

Results:

The results section is clear and well-organized.

Discussion:

5. The discussion is well written. However, authors are suggested to make connection between the observed changes in skeletal muscle strength and the potential implications for the development of type 2 Diabetes mellitus.

6. Authors should add and discuss the limitations of the study.

General Comments:

1. Authors should ensure consistent terminology throughout the manuscript. e.g.  Abbreviation "ND" is used instead of "NPD" in some instances.

2. Authors should re-check the manuscript for grammatical and typographical errors.

3. Authors should add abbreviations list.

Comments on the Quality of English Language

Authors should re-check the manuscript for grammatical and typographical errors.

Reviewer 2 Report

Comments and Suggestions for Authors

Dear Editors,

The paper "Investigating the Effects of Diet-Induced Prediabetes on Skeletal Muscle Structure and Muscle Strength in Male Sprague Dawley Rats" by Mandlakazi Dlamini and Andile Khathi explores the impact of a high-fat, high-carbohydrate diet on the development of prediabetes and its subsequent effect on skeletal muscle structure and function in male Sprague Dawley rats. This adds valuable insights into the pathophysiological changes preceding Type 2 Diabetes Mellitus. However, there are some drawbacks and concerns with lacking experiments for a more comprehensive understanding of the topic.

Major

- With only 12 rats, divided into two groups, the small sample size could limit the generalizability of the findings and increase the risk of Type II errors. Expanding the sample size could enhance the robustness of the results. If this is not possible, the authors are advised to at least present their data as histograms with individual scatter points showing the raw data.

- A major drawback of the study lies in the absence of mechanistic studies. Incorporating analyses of muscle fiber type composition, satellite cell activation, and inflammatory markers offers mechanistic insights into how prediabetes affects muscle health. Besides, authors are advised to conduct histological studies on the muscle tissue from the respective animal groups. Nevertheless, they mention that gastrocnemius skeletal muscles were harvested, but no results are shown.

-  While the study mentions the use of a standard rat chow for the control group, it does not specify its composition. A detailed comparison of the nutritional profiles of the control and HFHC diets would better contextualize the dietary impact.

- The authors mention the oral glucose tolerance test in the methods section but fail to show any related results. This is required for the paper.

- Did the data follow a normal distribution? How did the authors check for it? If normality is not met, Mann-Whitney should be used instead.

- The discussion is well-written and provides a thoughtful interpretation of the findings. However, it remains speculative regarding the outcomes presented, underscoring the need for mechanistic investigations to substantiate the observed effects.

Minor

- What is the rationale for using only male rats?

Reviewer 3 Report

Comments and Suggestions for Authors

This paper is a study evaluating the effect of diabetes and muscle strength. For this purpose, the comparison group was divided into pre-diabetic and non-diabetic groups through measurement of glycated hemoglobin concentration and fasting blood glucose concentration, and a correlation was proven through a grip strength test.

Understandably, the current experimental results prove that the skeletal muscles are affected in the pre-diabetic group, but blood and tissues were collected and biochemically analyzed, but the results need to be supplemented by adding tables and figures.

Additional comments:

The following information must be corrected and supplemented.  

- Figure (a) in Figure 2 is only a part, so I would like to change it to the entire experiment environment.  

- Understandably, the current experimental results prove that the skeletal muscles are affected in the pre-diabetic group, but, in 4.5 addressed blood and tissues were collected and biochemically analyzed, but the results need to be supplemented by adding tables and figures.  

- The experimental results on the effect of diabetes on the musculoskeletal system and muscle strength are important, but there is a need to include and explain the chemical analysis results.

I hope you will conduct a comparison in Section 4.8 and add more explanation than statistical significance.

Round 2

Reviewer 2 Report

Comments and Suggestions for Authors

Dear Editors,

The authors have addressed all of the concerns, with the exception of the mechanistic research, due to logistical problems and a shortage of lab supplies. The authors are therefore encouraged to include a limitation paragraph before the conclusion.

Regards
